# The Gut Microbiome Regulates the Psychomotor Effects and Context-Dependent Rewarding Responses to Cocaine in Germ-Free and Antibiotic-Treated Animal Models

**DOI:** 10.3390/microorganisms13010077

**Published:** 2025-01-03

**Authors:** Andrew D. Winters, Dina M. Francescutti, David J. Kracht, Diptaraj S. Chaudhari, Branislava Zagorac, Mariana Angoa-Perez

**Affiliations:** 1Department of Physiology, School of Medicine, Wayne State University, Detroit, MI 48201, USA; 2John D. Dingell VA Medical Center, Detroit, MI 48201, USA

**Keywords:** germ-free, antibiotics, gut microbiota, cocaine

## Abstract

Cocaine use disorder remains a major global health concern, with growing evidence that the gut microbiome modulates drug-related behaviors. This study examines the microbiome’s role in cocaine-induced psychomotor activation and context-dependent reward responses using germ-free (GF) and antibiotic-treated (ABX) models. In GF mice, the absence of a microbiome blunted cocaine-induced psychomotor activation (*p* = 0.013), which was restored after conventionalization. GF mice also showed reduced cocaine-conditioned place preference (CPP) (*p* = 0.002), which normalized after conventionalization. Dopaminergic function, critical for psychomotor responses and reward, was microbiome-dependent, with increased dopamine levels (*p* = 0.009) and normalized turnover ratios after conventionalization. In the ABX model, microbiome depletion reduced both cocaine-induced locomotion and CPP responses (*p* ≤ 0.009), further supporting the role of gut microbes in modulating psychomotor and reward behaviors. ABX-treated mice also showed significant declines in microbial diversity, shifts in bacterial structure, and dysregulation in metabolic, immune, and neurotransmitter pathways (*p* ≤ 0.0001), including alterations in short-chain fatty acids and gamma-aminobutyric acid metabolism. These findings highlight the gut microbiome’s critical role in regulating cocaine’s psychomotor and rewarding effects, offering insights into potential therapeutic strategies for cocaine use disorder.

## 1. Introduction

Cocaine abuse is one of the most prevalent drug use disorders and represents a significant and worsening global health challenge, exacerbated by increasing drug supply and consumption trends [1,2,3]. Cocaine’s addictive properties have long been attributed to its central nervous system effects, including enhanced dopamine transmission and alterations in reward-related circuitry. However, peripheral physiological effects of cocaine, such as dopamine release from peripheral terminals and activation of the peripheral nervous system, are increasingly recognized as key contributors to its addictive nature, functioning as conditioned stimuli that reinforce central drug effects [4]. Despite advances in understanding cocaine’s neurobiology, relapse rates remain alarmingly high, highlighting the need for a broader perspective on factors that modulate addiction and recovery.

Recent evidence highlights the critical role of the gut microbiome in modulating drug-related behaviors, including reward and relapse vulnerability. The gut–brain axis, a bidirectional communication system involving the nervous, endocrine, and immune systems, facilitates interactions between the gut microbiota and the central nervous system [5]. Microbial metabolites, such as short-chain fatty acids (SCFAs) like butyrate, influence neurobehavioral outcomes by modulating inflammation, epigenetic regulation, and neurotransmitter synthesis [5,6]. Dysbiosis, or microbial imbalance, has been implicated in stress sensitivity, anxiety, and addiction-like behaviors, further emphasizing the microbiome’s contribution to neuropsychiatric and addictive disorders [5,7].

In the context of cocaine, significant alterations in gut bacteria caused by antibiotics have been shown to increase cue-induced cocaine-seeking behavior following prolonged abstinence and to affect gene expression in brain regions crucial for reward processing [8]. These effects can be reversed with the supplementation of bacterial fermentation metabolites, such as butyrate, underscoring the importance of microbial products in modulating cocaine-related neurobehavioral responses [8,9]. Additionally, colonization with specific gamma-Proteobacteria members has been reported to enhance neurobehavioral responses to cocaine, possibly through microbial signaling pathways that influence dopamine transmission [10]. The gut microbiome’s role extends beyond antibiotics, as calorie-dense diets capable of altering the microbiota have also been shown to modify drug reward behaviors through overlapping neural circuits [6]. While much of the evidence relies on antibiotic manipulation, these approaches are limited by off-target effects and non-specific microbial depletion [11]. Germ-free (GF) models represent a complementary and valuable tool for studying host–microbial interactions, as they allow precise examination of the microbiome’s influence on drug-related behaviors without the confounding effects of antibiotics [12].

In this study, we aimed to explore psychomotor and context-dependent rewarding responses to cocaine using both germ-free (GF) and antibiotic-treated (ABX) models. By employing these complementary approaches, we sought to provide deeper insights into the role of the gut microbiome in modulating cocaine reward and relapse.

## 2. Materials and Methods

### 2.1. Animals Models

GF mice were tested both before and after conventionalization to assess the effects of introducing a microbiome. In contrast, the ABX model simulated microbiome depletion, providing a comparative framework to understand how varying degrees of microbiome alteration impact cocaine-dependent behaviors, neurochemistry, and metabolism. Cocaine hydrochloride was obtained from the NIDA Controlled Substances Program and administered by i.p. injection at a dose of 20 mg/kg (dissolved in sterile 0.9% physiological saline) as this has shown robust stimulatory effects on motor activity. All control mice received sterile saline injections. All animals had ad libitum access to food and water and received Diet 5013 (LabDiet, St. Louis, MO, USA), which is composed of 20% protein, 5% crude fat, 6% crude fiber, and 0.8% ash. For the GF model, all food was autoclaved. All procedures involving the use of animals in this study were reviewed and approved by the Wayne State University Institutional Animal Care and Use Committee (protocol # 22-02-4406). The number of mice included in each analysis is shown in Appendix A.

### 2.2. Model Design

Adult (7–8 weeks old) C57BL/6 mice (Inotiv, Indianapolis, IN, USA) were used in both the GF and ABX models. GF male and female mice were purchased from the University of Michigan Germ-Free Mouse Facility and transported in sterile GF shippers, while conventionally raised (CONV-R) C57BL/6 mice matched for age and sex were obtained from Charles River Labs for use as controls in the GF model. GF mice were housed in sterile cages and maintained in a biosafety cabinet when not undergoing behavioral testing. Gram stains, performed according to [13], and quantitative real-time PCR (qPCR), described below, of fecal samples confirmed the absence of a gut microbiome in GF mice upon arrival and the status of the gut throughout the conventionalization phase. For the GF model, conventionalization involved adding litter from CONV-R cages to GF cages every other day until day 27, when Gram stains and qPCR results indicated that bacterial loads were indistinguishable from controls. For the ABX model, CONV-R C57BL/6 mice were treated with an oral gavage of an antibiotic cocktail (ampicillin, neomycin, metronidazole at 100 mg/kg each, and vancomycin at 50 mg/kg) prepared in sterile water daily for 21 days, following a 7-day acclimation period to ensure uniform housing. Controls received sterile water by oral gavage. Post-treatment, animals were maintained under controlled conditions to minimize external variables.

### 2.3. Psychomotor Activation

Psychomotor activation was measured in 16 transparent plastic chambers (Omnitech Electronics LLC, Columbus, OH, USA). Immediately after cocaine administration, mice were placed in the center of the cage and allowed to move freely for 30 min. During that time, activity was measured by infrared light beam arrays in the horizontal and vertical axes. Motor activity was recorded on a computer and analyzed using Fusion software (v 6.5, Omnitech). Total activity was defined as the distance traveled and was represented by the sum of all beam breaks in both horizontal and vertical planes during the 30 min recording session.

### 2.4. Conditioned Place Preference (CPP)

This paradigm was tested as previously described by Manzanedo et al., 2020 [14], with slight modifications. In brief, eight identical Plexiglas boxes with two equally sized compartments separated by a central removable door (Omnitech Electronics LLC) were employed. The compartments consisted of different patterned walls (stripes vs. dots) and distinct floor textures (rubber non-skid vs. smooth surfaces). Photobeam sensors in each compartment of the box allowed the tracking of the animal, and a computer with the Fusion software recorded all sessions. For the initial pretest phase, the preference for each compartment was recorded for 20 min for each animal. During the acquisition phase, mice were conditioned by pairing cocaine administration with a 20-min session in which they were confined to the chamber opposite to their baseline preference. Two sessions (saline–saline for controls or saline–cocaine for treated mice) were conducted each day for 2 days. On the 4th day, CPP was tested by allowing animals free access to both sides and recording time spent in each side. The difference in seconds between the time in the cocaine-paired side in the test and the time spent in the same side in the pretest is considered a measure of the degree of drug-induced conditioning and was used to calculate a CPP score with percentage times.

### 2.5. Novel Object Recognition

This test, used to evaluate recognition memory based on the innate tendency of rodents to explore their environment, was performed to confirm that the exploratory behavior and cognitive function of GF mice were intact, thereby supporting the use of this model for investigating how alterations in the microbiome influence various behavioral outcomes. GF and CONV-R mice were tested in this paradigm according to previous studies [15]. Exploration of an object was defined as rearing on the object, sniffing it at a distance of less than 1 cm and/or touching it with the nose. Successful recognition was represented by preferential exploration of the novel object over the familiar object. The percentage of time spent exploring each object (novel vs. familiar) was plotted using the total time investigating to normalize measures among individuals.

### 2.6. Determination of Dopamine Transporter (DAT) and Tyrosine Hydroxylase (TH) Protein Levels

The effects of the lack of a gut microbiome on striatal DAT and TH levels were determined by immunoblotting as previously described [16]. In short, tissue was dissected from the brain and stored at −80 °C. Frozen tissue was sonicated in 1% SDS at 95 °C, and insoluble material was removed. Soluble protein concentrations were determined by the bicinchoninic acid method. Equal amounts of protein were resolved by SDS-polyacrylamide gel electrophoresis and then electroblotted to nitrocellulose. Blots were blocked in Odyssey blocking buffer for 1 h at room temperature. Primary antibodies against DAT (Abcam 1:1000), TH (in-house antibody 1:1000), or GAPDH (Sigma 1:10,000) were added and incubated overnight at 4 °C. Blots were washed in Tris-buffered saline and then incubated with IRDye secondary antibodies (1:4000) for 1 h at room temperature. Immunoreactive bands were visualized by enhanced fluorescence, and the relative densities of TH-, DAT-, and GAPDH-reactive bands were determined by imaging with an Odyssey CLx Infrared Image System (LiCor Biosciences, Lincoln, NE, USA) and quantified using Image Studio software (v 5.0, LiCor). DAT and TH relative densities were normalized to the GAPDH level for each lane to control for loading errors.

### 2.7. Determination of Dopamine (DA) Level and Turnover Rate

The ventral striatum was dissected from the brains of mice and stored at −80 °C. Frozen tissues were weighed and sonicated in 10 vol of 0.16 N perchloric acid at 4 °C. Insoluble protein was removed by centrifugation, and DA and 3,4-dihydroxyphenylacetic acid (DOPAC) were determined by HPLC with electrochemical detection according to Anneken et al., 2018 [16]. Briefly, supernatant diluted 1:8 in 0.16 N perchloric acid was injected via autosampler onto a C-18 reverse phase column in buffer (100 mM citric acid, 75 mM NaH_2_PO_4_, 176 mg/L octanesulfonic acid, 50 mg/L EDTA, 16.5% methanol, pH 4.5). Peaks were quantified on the basis of known standards and corrected for dilution. DA turnover rate was calculated by dividing the levels of DOPAC by DA.

### 2.8. Bacterial Load

Assessment of bacterial load, defined here as bacterial 16S rRNA gene copy number per mg of feces, was assessed via amplification of the V4 region of the 16S rRNA gene using qPCR, as previously described [17]. Briefly, each reaction consisted of PowerUp™ SYBR™ Green Master Mix (Applied Biosystems, Waltham, MA, USA), primers 515F and 806R, nuclease-free water, and purified DNA or water. The qPCR was performed using the following cycling conditions: 95 °C for 2 min, followed by 37 cycles of 95 °C for 30 s, 57 °C for 30 s, and 72 °C for 30 s. All qPCRs and data collection were carried out with an ABI QuantStudio 3™ real-time PCR system using software v2.3 (Applied Biosystems, Waltham, MA, USA). Raw amplification data were analyzed using the online platform Thermo Fisher Cloud, Standard Curve Analysis Module online platform (version 3.9) with an automatic baseline setting and a fluorescence threshold of 0.25. Reactions were performed in duplicate, and the average Cq values were determined based on these duplicates, representing the cycle number required for exponential increases in fluorescence. The copy number of the 16S rRNA gene in samples was then calculated as described by Gallup, 2001 [18] using the equation X_σ_= E_AMP_^b−Cq^. E_AMP_ represents the exponential amplification value for the assay, which is represented as E_AMP_ = 10^−1/*m*^, while *m* and *b* represent the slope and intercept of the above regression. A synthesized gene block served as the quantitation standard. Standard curves were included in duplicate for each run, and the copy numbers were log10 transformed for normalization before analysis. The rstatix package version 0.7.2 [19] was used to detect outliers.

### 2.9. 16S rRNA Gene Amplicon Sequencing and Bioinformatic Processing

The V4 region of the 16S rRNA gene was amplified and sequenced using the dual indexing strategy developed by Kozich et al., 2013 [20], as previously employed in our laboratory [17,21]. Sequence read files were then processed using the Divisive Amplicon Denoising Algorithm, DADA2 [22] pipeline, to produce an amplicon sequence variant (ASV) count table. Forward and reverse reads were truncated at 155 and 155 bases, respectively. Sequences were then classified using the silva_nr_v138_train_set database with a minimum bootstrap value of 80%, and a single ASV that was derived from mitochondria and Chloroplast were removed.

### 2.10. Inference of Functional Genes and Pathways

PICRUSt2 software package version 2.5.2 [23] was used to predict functional pathway occurrence based on marker gene sequences (16S rRNA sequencing data). MetaCyc ontology predictions [24] were used for metabolic pathways classification.

### 2.11. Statistical Analyses

Initially, sex effects were tested using two-way ANOVA for each behavioral task and neurochemistry. When no differences were found, groups were collapsed and re-analyzed to include microbial status in the comparisons. Two-way ANOVA with subsequent Tukey’s post hoc tests were performed to analyze the data from behavioral tests as well as the neurochemistry data from HPLC, using Prism (GraphPad v 10). Immunoblotting was analyzed with unpaired *t*-tests.

For both the GF and ABX models, alpha diversity (Chao1, Shannon, and Inverse Simpson) were calculated in R using the phyloseq function estimate_richness (phyloseq version 1.44.0; [25]).

For the GF model, prior to the analyses of alpha diversity metrics, the 16S rRNA gene profiles were subsampled to a sequencing depth of 38,967 reads per sample ‘rarefy_even_depths’ from phyloseq. The random seed value was set to 1, and random sampling was performed without replacement. For the ABX model, the 16S rRNA gene profiles were subsampled to a sequencing depth of 10,572 prior to analyses. Variation in alpha diversity metrics of bacterial profiles was assessed using two-way ANOVA models (lmer). ANOVA (car R package; [26]) was utilized to evaluate each diversity metric for statistical significance, followed by Tukey’s pairwise comparisons (emmeans package) to detect significant differences between experimental groups. For all analyses, fixed effects were assessed using Type II Wald F tests, and interactions were assessed using Type II Wald F tests.

Variation in beta diversity of bacterial profiles was characterized using the Bray–Curtis similarity index as implemented in the R package vegan version 2.6.4 [27] on count data and visualized using principal coordinates analysis (PCoA) plots. Differences in the bacterial community structure between the fecal samples of C57BL/6 and GF mice were evaluated using permutational multivariate analysis of variance (PERMANOVA) as implemented in vegan version 2.6.4.

Differential relative abundance of bacterial phyla and metabolic pathways was assessed using a linear mixed-effects model as implemented in the R package MaAsLin2 version 1.14.1 [28]. The model settings included min_prevalence = 0.25, normalization = TMM, transform = NONE, and analysis_method = NEGBIN. The resulting *p*-values were adjusted for multiple comparisons using the Benjamini–Hochberg method [29].

## 3. Results

### 3.1. Psychomotor Activation

Cocaine treatment induced psychomotor activation, evidenced by increased locomotion in both the GF and ABX mouse models (Figure 1A–C). In the GF model, CONV-R mice treated with cocaine traveled a significantly greater distance than saline-treated counterparts (*p* = 0.0001, Tukey’s *p* = 0.0004) (Figure 1A). In contrast, GF mice exhibited a blunted locomotor response to cocaine, which, while elevated compared to saline-treated GF mice, did not reach statistical significance (*p* > 0.05). GF mice treated with cocaine also showed significantly reduced locomotion compared to CONV-R mice treated with cocaine (*p* = 0.0132, Tukey’s *p* = 0.0134). No significant differences were observed between saline-treated CONV-R and GF mice (*p* = 0.9834) or between cocaine- and saline-treated GF mice (*p* = 0.4383). The interaction between cocaine treatment and GF status was significant (*p* = 0.0424), suggesting that the absence of gut microbiota attenuates cocaine-evoked psychomotor activation. Locomotor responses did not differ by sex (*p* > 0.05), allowing data to be collapsed across sexes.

Restoration of the gut microbiota in GF mice normalized cocaine-evoked psychomotor activation (Figure 1B). As expected, CONV-R mice treated with cocaine showed significantly increased locomotion compared to saline controls (*p* = 0.0008). Additionally, CONV-GF mice also exhibited heightened locomotion with cocaine compared to their counterparts treated with saline (*p* = 0.0019). Moreover, there was no significant difference between cocaine-treated CONV-R and CONV-GF mice (*p* = 0.9927), indicating restitution of normal psychomotor responses. In this conventionalization phase, sex effects remained non-significant (*p* > 0.05), and neither GF status nor the interaction between GF status and cocaine treatment was significant (*p* = 0.8436 and *p* = 0.8511, respectively).

In the ABX model, although cocaine increased locomotion, this effect was attenuated (Figure 1C). Before treatment with antibiotics, cocaine induced a robust increase in total distance traveled (*p* < 0.0001), and no sex effects were observed, allowing data to be collapsed across sexes. Post-antibiotic analysis revealed significant main effects of cocaine (*p* < 0.0001) and antibiotics (*p* = 0.0014), as well as a significant interaction between the two (*p* = 0.0263). Cocaine-treated control mice traveled significantly more than both saline-treated controls (*p* < 0.0001) and saline-treated ABX mice (*p* < 0.0001). While cocaine-treated ABX mice also traveled more than saline control mice (*p* = 0.0157), their locomotion was significantly reduced (*p* = 0.001). No significant differences were observed between the two saline-treated groups (*p* = 0.8707).

Together, these findings demonstrate that both the absence of a gut microbiota and antibiotic-mediated disruption attenuate cocaine-evoked psychomotor activation, though the extent of attenuation differs between models. Restoration of the gut microbiota in GF mice normalized locomotor responses to cocaine, whereas antibiotics partially inhibited cocaine-induced locomotion. These results emphasize the critical role of the gut microbiome in modulating psychomotor responses to cocaine.

### 3.2. CPP

The acquisition of cocaine CPP differed between GF and ABX mouse models (Figure 1D,E). In the GF model, CONV-R mice displayed a robust CPP, whereas GF mice exhibited a blunted response (Figure 1D). Analysis showed a significant effect of conventionalization (*p* = 0.0458) and its interaction with GF status (*p* = 0.0342), but not GF status alone (*p* = 0.1096). Post hoc analysis indicated that, while, at the beginning of theisstudy, CONV-R mice had significantly higher baseline CPP scores compared to GF mice (*p* = 0.0021), this difference was no longer observed following conventionalization, as CPP scores of GF mice were comparable to those of CONV-R mice post-conventionalization (*p* > 0.5).

In the ABX model, significant effects of antibiotic treatment on cocaine CPP scores were observed (*p* = 0.0020), with ABX mice exhibiting reduced CPP compared to controls (Figure 1E). Neither sex (*p* = 0.2500) nor the interaction between antibiotic treatment and sex (*p* = 0.1909) had significant effects, allowing data to be collapsed across sexes. Post hoc comparisons of cocaine-treated mice revealed that control males had higher CPP values compared to ABX males (*p* = 0.0061). While control females also had higher predicted mean CPP scores compared to ABX females, this difference did not reach statistical significance (*p* > 0.05).

These findings suggest that both the absence of gut microbiota and antibiotic treatment impair the acquisition of cocaine CPP. In GF mice, the normalization of the gut microbiota through conventionalization restored CPP acquisition, underscoring the critical role of the gut microbiome in modulating conditioned responses to cocaine.

### 3.3. NOR

To rule out differences in GF mice that would influence their performance in CPP, an evaluation for exploratory behaviors and cognitive function was performed using the NOR test. Results indicate that exploratory behaviors in GF mice are normal, and no cognitive impairments are present. Both GF and CONV-R mice exhibited similar time investigating a familiar object during the NOR test (Figure 1F). Importantly, there was a significant increase in the time both GF and CONV-R mice spent exploring the novel object compared to the familiar object (*p* < 0.0001), regardless of GF status. These findings demonstrate that gut GF status does not influence exploratory behaviors or cognitive performance.

### 3.4. Striatal Levels of DAT and TH

Microbial status did not significantly affect striatal DAT and TH levels in either the GF or ABX models (Figure 2A–D). In the GF model, no significant differences were observed for DAT levels based on GF status (*p* = 0.3859), GF status (*p* = 0.6974), or the interaction of the two effects (*p* = 0.8832) (Figure 2A).

Similarly, in the ABX model, there were no significant differences in striatal DAT levels between control and antibiotic-treated mice (*p* = 0.8098). When examining the influence of sex on DAT levels in the ABX model, splitting by sex revealed a significant effect of sex (*p* < 0.0001), with females exhibiting higher DAT levels compared to males in both control and ABX groups (Figure 2B). Although the main effect of ABX was not significant (*p* = 0.0669), post hoc comparisons showed significant differences between male and female groups (*p* < 0.05 for male control vs. female control and *p* < 0.001 for male ABX vs. female ABX). These results suggest that sex significantly influences DAT levels, with females showing higher levels than males in both conditions.

Additionally, in the GF model, there were no significant differences in TH levels between the CONV-R and GF mice (*p* = 0.9284), between pre-CONV and CONV groups, (*p* = 0.6919), or the interaction of the two effects (*p* = 0.685) (Figure 2C). In the ABX model, sex significantly influenced TH levels (*p* = 0.013), with females having higher TH levels than males. There was no significant effect of antibiotic treatment (*p* = 0.5204), and the interaction between sex and antibiotic treatment was not significant (*p* = 0.7922) (Figure 2D).

### 3.5. Striatal DA Levels and Turnover Rate

In both the GF and ABX models, microbiome manipulation had differential effects on DA levels in the striatum (Figure 3A,B). In the GF model, the transition to a conventional microbiome in GF mice significantly influenced DA levels, with a marked increase observed between pre-CONV and CONV (*p* = 0.009). However, DA levels in the CONV-R group remained stable, showing no significant differences across both time points (*p* > 0.05) (Figure 3A). In contrast, in the ABX model, no significant difference in DA levels was observed between the control and antibiotic-treated groups (*p* = 0.371) (Figure 3B).

Microbial manipulation significantly influenced DOPAC levels in the GF model but not in the ABX model (Figure 3C,D). In the GF model, two-way ANOVA revealed significant effects of GF status (*p* < 0.0001) and conventionalization (CONV) (*p* = 0.0021) (Figure 3C). Specifically, pre-CONV DOPAC levels in GF mice were significantly greater than those in CONV-R mice (*p* < 0.0001). After conventionalization, DOPAC levels in GF mice significantly decreased (*p* < 0.0001) and became comparable to those in CONV-R mice (*p* > 0.05). Additionally, the two-way ANOVA revealed significant effects of sex (*p* < 0.0001), antibiotics (*p* = 0.0057), and their interaction (*p* = 0.0033) on DOPAC levels in the striatum. All possible pairwise comparisons except for male control vs. male ABX) significantly differed (Figure 3D).

Microbiome manipulation significantly influenced the DOPAC/DA ratio in both models (Figure 3E–G). In the GF model, the DOPAC/DA ratio at pre-CONV was significantly greater in GF mice compared to CONV-R mice (*p* < 0.0001) (Figure 3E). However, following conventionalization, the ratio in GF mice significantly decreased (*p* < 0.0001) and became comparable to that in CONV-R mice (*p* > 0.05). Sex significantly affected the DOPAC/DA ratio in the GF model, with females having higher DOPAC/DA ratios than males (*p* = < 0.0001). However, a decrease in the DOPAC/DA ratio in CONV mice compared to pre-CONV mice was observed for both males and females (*p* < 0.0001) (Figure 3F).

In contrast, in the ABX model, no significant differences in DOPAC/DA ratios were observed between control and ABX mice when sex data were merged (*p* = 0.6082). When considering sex separately, females exhibited consistently higher DOPAC/DA turnover levels than males, with significant differences observed between males and females in both control and antibiotic-treated groups (*p* < 0.0001) (Figure 3G).

### 3.6. Bacterial Load

Analysis of bacterial load revealed no significant effects in either the GF or ABX model (Figure 4A,G). In the GF model, there was no significant effect of GF status (pre-CONV vs. post-CONV) (*p* = 0.079), sex (*p* = 0.959), or the interaction between GF status and sex (*p* = 0.944), indicating a return to a conventional gut microbiome status following conventionalization in GF mice (Figure 4A).

Similarly, in the ABX model, qPCR analysis of bacterial load after 27 days revealed no significant effect of treatment, sex, or the interaction of the two effects (*p* > 0.05), indicating that conventionalization of the gut with antibiotic-resistant bacteria occurred without any significant differences across treatment groups (Figure 4G).

### 3.7. Microbial Alpha and Beta Diversity

In the GF model, significant differences in alpha diversity were observed across treatment groups, with both GF status and sex influencing the results (Figure 4B–D,H–J). In the GF model, both initial microbiome status (*p* = 2.6 × 10^−11^) and sex (*p* = 1.0 × 10^−8^) significantly influenced Chao1 index values (Figure 4B), with females showing greater richness than males (*p* < 0.0001) in the CONV group. In contrast, no significant difference in richness was observed between male and female GF mice (*p* = 1.0000), and there was no significant interaction between sex and GF status (*p* = 0.617). For the Shannon index (Figure 4C), GF status had a significant effect (*p* = 0.00041), with CONV mice exhibiting greater diversity than GF mice, but sex did not have a significant impact (*p* = 0.43183). The interaction between sex and GF status was not significant (*p* = 0.24272). In the analysis of the inverse Simpson index (Figure 4D), initial microbiome status had a significant effect in the GF model (*p* = 0.00073). Sex also had a significant effect (*p* = 0.02569), and the interaction between sex and GF status was significant (*p* = 0.04699). Pairwise comparisons indicated that female conventionalized mice had significantly higher inverse Simpson diversity compared to female GF mice (*p* = 0.0037).

In the ABX model, significant differences in alpha diversity were also observed (Figure 4H–J). For the Chao1 index, significant effects of initial microbiome status (*p* = 4.0 × 10^−11^), sex (*p* = 5.3 × 10^−5^), and the interaction of the two effects were observed (Figure 4H). Females showed significantly greater richness than males in the control group (*p* = 0.0003), while there was no significant difference in richness between male and female ABX mice (*p* = 0.8995). For the Shannon index (Figure 4I), the initial microbiome status showed a significant effect (*p* < 2 × 10^−16^), while sex (*p* = 0.078) and the interaction between sex and antibiotic treatment (*p* = 0.063) did not reach statistical significance. The Shannon diversity was significantly higher in the control group compared to the ABX group. For the Inverse Simpson index (Figure 4J), significant effects of initial microbiome status, sex, and the interaction of the two effects (*p* < 0.0001) were observed. Females showed significantly greater diversity than males in both the control and ABX groups (*p* < 0.0001). The only exception was the comparison between male and female ABX mice, where no significant difference was observed (*p* = 0.9988).

At the end of this study, after conventionalization of the GF mice, beta diversity analysis using Bray–Curtis distances revealed significant effects of initial GF status (*p* = 1 × 10^−4^), sex (*p* = 1 × 10^−4^), and the interaction of the two effects (*p* = 1 × 10^−4^). CONV-R male mice did not significantly differ from GF male mice (*p* = 0.6000); however, GF females differed from GF males (*p* = 0.0360). CONV-R female mice differed from GF female mice (*p* = 0.0012), CONV-R male mice (*p* = 0.0252), and GF male mice (*p* = 0.0294). CONV-R male mice also significantly differed from GF female mice (*p* = 0.0258). The PCoA plot (Figure 4E) illustrates the separation of the bacterial community profiles by sex and GF status. (Figure 4E).

Beta diversity analysis in the ABX mice revealed significant effects of both treatment (*p* = 1 × 10^−4^) and sex (*p* = 7 × 10^−4^), as well as a significant interaction between treatment and sex (*p* = 9 × 10^−4^). Except for the comparison between ABX females and males (*p* = 0.0757), all other pairwise comparisons between treatment and sex were significant (*p* < 0.0018). The PCoA plot (Figure 4K) illustrates that male and female ABX mouse samples completely overlap, highlighting the lack of a significant difference between these groups (Figure 4K). This suggests that antibiotic treatment caused a substantial shift in community structure, with a distinct separation based on sex, while the overlap between males and females further suggests limited variation within treatment groups.

### 3.8. Differential Abundance

In the comparison between GF and control mice, significant shifts in microbial community structure were observed (Figure 4F,L and Figure 5A,B). Several Lachnospiraceae ASVs were notably reduced in GF mice, with fold differences reaching as low as −11.101 (*p* < 0.0001). Conversely, ASVs from *Akkermansia* and Muribaculaceae were more abundant in GF mice, including *Akkermansia* ASV281 (fold change = 5.874), *Akkermansia* ASV269 (fold change = 5.008), *Akkermansia* ASV238 (fold change = 4.876), *Akkermansia* ASV7 (fold change = 4.960), Muribaculaceae ASV22 (fold change = 8.705), and Muribaculaceae ASV132 (fold change = 6.685) (*p* < 0.001).

In the analysis of differential abundance between male and female GF mice, 22 Muribaculaceae ASVs were found to be more abundant in male mice, with fold differences ranging from 0.922 to 10.003. For example, Muribaculaceae ASV22 had a fold difference of 10.003 (q = 2.23 × 10^−7^), Muribaculaceae ASV45 had a fold difference of 9.482 (q = 9.25 × 10^−84^), and Muribaculaceae ASV159 had a fold difference of 9.421 (q = 1.19 × 10^−54^). Several Lachnospiraceae ASVs exhibited a pronounced pattern, with 27 identified ASVs being more abundant in male mice. Key ASVs Lachnospiraceae ASVs in male mice included ASV319 (fold difference: 8.215, q = 3.77 × 10^−9^), ASV57 (fold difference: 8.157, q = 2.35 × 10^−11^), and ASV177 (fold difference: 5.972, q = 1.35 × 10^−7^).

In the differential ASV abundance analysis, antibiotic treatment led to significant changes in microbial community structure. Several ASVs from the Lachnospiraceae family were notably reduced in antibiotic-treated mice, with fold changes ranging from −4.91 (q = 0.000306061) to −7.86 (q = 4.16 × 10^−6^). Conversely, ASVs from the Enterobacteriaceae family increased significantly, including Enterobacteriaceae ASV1 (fold change = 15.96, q = 9.11 × 10^−123^) and *Proteus* ASV70 (fold change = 15.68, q = 1.63 × 10^−283^). Other notable changes include increases in *Enterococcus* ASV4 (fold change = 12.36, q = 5.15 × 10^−31^) and decreases in *Alistipes* ASV54 (fold change = −7.59, q = 2.96 × 10^−9^) and Candidatus Arthromitus ASV30 (fold change = −7.49, q = 5.13 × 10^−12^). Additionally, several ASVs from the Muribaculaceae family were significantly reduced, including Muribaculaceae ASV115 (fold change = −6.63, q = 5.39 × 10^−9^), Muribaculaceae ASV119 (fold change = −6.72, q = 5.10 × 10^−10^), and Muribaculaceae ASV134 (fold change = −7.24, q = 7.42 × 10^−10^).

Regarding sex differences, in the ABX model, several Lachnospiraceae ASVs were significantly lower in males, including ASV430 (fold change = −1.11, q = 7.97 × 10^−87^), ASV40 (fold change = −2.38, q = 8.83 × 10^−5^), and ASV19 (fold change = −4.47, q = 0.000258465). One Lachnospiraceae ASV, ASV57, was more abundant in males (fold change = 4.87, q = 0.004795298). Conversely, several Muribaculaceae ASVs were less abundant in males compared to females, including ASV134 (fold change = −4.64, q = 0.000112572), ASV68 (fold change = −5.19, q = 8.51 × 10^−5^), ASV20 (fold change = −5.25, q = 4.12 × 10^−5^), and ASV53 (fold change = −4.05, q = 0.000306061). Two Muribaculaceae ASVs were more abundant in males, including ASV121 (fold change = 1.97, q = 1.58 × 10^−7^) and ASV22 (fold change = 0.46, q = 1.38 × 10^−138^).

### 3.9. Inferred Microbial Metabolic Pathways Abundance

The analysis of inferred microbial pathways from fecal samples revealed more pronounced metabolic differences in the ABX model compared to the GF model (Figure 6A–D). In contrast to the relatively minor metabolic changes observed in conventionalized GF mice, which showed a few downregulated pathways (e.g., methanogenesis from acetate, L-lysine fermentation, and cob(II)yrinate ac diamide biosynthesis I) relative to baseline GF mice (Figure 6A,B), ABX mice exhibited broader shifts across multiple functional groups (Figure 6C,D).

In ABX mice, amino acid metabolism was notably altered, with increased degradation of arginine and ornithine, as seen in pathways such as L-arginine degradation II, putrescine production, and gamma-aminobutyric acid (GABA) degradation. These changes indicate substantial shifts in nitrogen metabolism, with increased GABA degradation suggesting potential effects on neurotransmitter levels and behavior.

Additionally, carbohydrate metabolism in antibiotic-treated mice favored alternative sugar pathways like sulfoglycolysis, while classical glycolytic routes were downregulated. In lipid and fatty acid metabolism, upregulation of biosynthetic and oxidative pathways (e.g., fatty acid beta-oxidation I) was observed, alongside a reduction in acetyl-CoA fermentation to butanoate and alterations in ubiquinol biosynthesis. These changes reflect a shift in energy demands and a potential decrease in butanoate (butyrate) production, which could influence SCFA levels.

The TCA cycle and glyoxylate cycle in ABX mice were upregulated, underscoring a shift toward aerobic metabolism, with concurrent reductions in fermentation and sulfur oxidation. Secondary metabolism in these mice showed increased biosynthesis of siderophores (e.g., aerobactin, enterobactin), suggesting adaptations for enhanced iron acquisition and immune evasion. Furthermore, cell wall biosynthesis favored lipid A and mycolate pathways over teichoic acid synthesis, indicating structural adjustments.

In contrast, the limited changes seen in the GF model suggest that the conventionalization process primarily results in minor metabolic shifts, whereas antibiotic treatment induces more substantial alterations in microbial community activity, modifying a wide range of metabolic pathways. These collective changes in ABX mice highlight an adaptive microbial response to antibiotic stress, with a focus on aerobic, oxidative energy metabolism, stress resilience, and altered carbohydrate and lipid utilization, potentially leading to significant changes in SCFA production.

## 4. Discussion

### 4.1. Findings of the Current Study

In the GF model, conventionalization of GF mice restored cocaine-induced psychomotor activation, which was absent in GF mice. Additionally, CONV-R mice displayed a robust CPP, whereas GF mice exhibited a blunted response, which was restored following conventionalization. Furthermore, DA levels were significantly elevated following microbiome acquisition, while baseline DOPAC levels were higher in GF mice compared to CONV-R mice. Upon conventionalization, DOPAC/DA ratios in GF mice decreased to levels seen in CONV-R mice, indicating that the microbiome influences DA metabolism. However, DAT and TH protein levels were unaffected by microbiome status. Microbial diversity was influenced by sex and microbiome status, with female CONV-R mice showing higher diversity than their GF counterparts. Despite these alterations, cognitive behavior, assessed via the NOR test, was not impaired (Figure 1F). Metabolic pathway predictions showed similar functional potential between the microbiomes of conventional and colonized GF mice.

In the ABX model, antibiotic treatment attenuated cocaine-induced locomotion, again suggesting that the microbiome influences psychomotor responses to the drug. Cocaine CPP scores were significantly reduced in ABX mice, further indicating the microbiome’s role in cocaine reward. Despite these behavioral changes, DA production in the striatum remained unaffected by antibiotic treatment, and no significant changes were observed in DAT or TH levels. DOPAC levels were higher in females, but antibiotic treatment did not influence DOPAC levels or DOPAC/DA ratios. Microbial diversity in antibiotic-treated mice was significantly lower than in controls, with decreased richness and distinct shifts in bacterial community structure and metabolic pathways, including dysregulation of metabolic and immune-related pathways.

### 4.2. Findings of the Current Study in the Context of Previous Studies

GF and ABX mouse models are both valuable tools for studying the gut microbiota’s role in health conditions like addiction [12,30,31], with each offering distinct advantages and limitations. GF mice, maintained in sterile environments, allow researchers to examine the complete absence of microbiota or specific microbial colonization, providing a clear and controlled model, though they require specialized facilities, high maintenance, and careful monitoring for contamination. Additionally, GF mice exhibit intact cognitive and exploratory behaviors, as demonstrated by the NOR test, validating their use in behavioral studies. In contrast, ABX mice offer a more accessible and flexible model for studying microbial depletion, although antibiotics may not fully eliminate the microbiota and can influence other physiological processes. In the current study, employing both models together, we investigated the effects of both complete microbiota absence and partial depletion, offering complementary insights into how microbiota alterations influence addiction-related behaviors and neurobiological mechanisms, thereby providing a more comprehensive understanding than either model alone.

### 4.3. Psychomotor Activation

The current study reveals that the gut microbiome plays a crucial role in modulating locomotor responses to cocaine, with similar effects observed in GF and ABX models. A unique contribution of this study is the restoration of cocaine-induced locomotion in GF mice after microbiome acquisition, a dynamic not explored in previous studies. GF mice exhibited reduced locomotion compared to CONV-R mice prior to microbiome acquisition, but this effect was reversed following conventionalization (Figure 1A,B). Similarly, antibiotic treatment attenuated cocaine-induced locomotion, with treated mice showing reduced activity compared to controls (Figure 1C). These findings highlight the importance of microbiota presence and composition in influencing psychomotor behavior.

In contrast, Kiraly et al. (2016) [9] and Cuesta et al. (2022) [10] reported enhanced cocaine-induced locomotion following microbiome disruption with antibiotics, suggesting a sensitizing effect in their models. However, their studies differ in methodology. Kiraly et al. (2016) [9] used a cocktail of neomycin, bacitracin, and vancomycin in male mice, observing significant locomotor effects at both 10 mg/kg and 5 mg/kg doses. Cuesta et al. (2022) [10] also used a cocktail of ampicillin, vancomycin, metronidazole, and neomycin in male mice, administering 5 mg of each antibiotic per mouse per day. Antibiotic treatment in their study reduced gut microbiota diversity and promoted colonization by γ-Proteobacteria, such as *Citrobacter rodentium* and *Escherichia coli*, which enhanced behavioral responses to cocaine. In contrast, Tran et al. (2023) [32] reported sex- and strain-specific increases in locomotor sensitization following antibiotic treatment in male and female C57BL/6J mice (CC004/TauUncJ and CC041/TauUncJ strains). The authors used a cocktail of sulfatrim, ampicillin, metronidazole, vancomycin, and neomycin in drinking water and observed that strain CC04, a high responder to cocaine sensitization, had a gut microbiome enriched with *Lactobacillus* and increased in *Barnesiella* following cocaine exposure. In contrast, strain CC041, characterized by a gut microbiome dominated by *Eisenbergella*, *Robinsonella*, and *Ruminococcus*, showed no significant microbiome changes in response to cocaine. These findings highlight the interplay between microbiome composition and its dynamic response to cocaine in driving divergent behavioral outcomes. Interestingly, sex-specific behavioral responses to antibiotics were only observed in one of the two strains tested. These differences illustrate the complexities of microbiome–behavior interactions, where experimental conditions, antibiotic regimens, sex, strain variability, and microbiome composition may result in divergent outcomes in locomotor response to cocaine.

### 4.4. CPP

In terms of CPP responses and the gut microbiome, in the current study, both the GF and ABX models showed that microbiome depletion significantly reduced CPP responses to cocaine. GF mice, which initially exhibited lower CPP scores, displayed restored CPP responses after conventionalization with a microbiota, while antibiotic-treated mice showed attenuated CPP responses, especially in males (Figure 1D,E). These results suggest that the presence or absence of a microbiome modulates cocaine CPP. These results are in line with established research on sex differences in cocaine addiction, where women tend to progress more quickly from initial use to dependence, report stronger euphoria from cocaine, and have higher rates of relapse compared to men [33].

These findings contrast with those of Kiraly et al. (2016) [9], who reported that antibiotic treatment enhanced CPP in mice at the lower 5 mg/kg dose of cocaine, where antibiotic-treated mice showed a robust place preference, while no effect above and beyond controls was observed at the higher 10 mg/kg dose. This discrepancy may be due to differences in antibiotic regimens, doses, or experimental designs. Kiraly et al. used a cocktail of non-absorbable antibiotics that reduced gut bacterial load, but 16S rRNA sequencing was not performed, leaving the effect on specific microbial communities unexplored. In contrast, Cuesta et al. (2022) [10] found that antibiotics disrupted the gut microbiota, reducing its diversity and enabling colonization by *C*. *rodentium* or *E*. *coli*, which enhanced cocaine-induced CPP responses. However, the effect of antibiotics alone was not investigated. Nonetheless, this shift in microbial composition, linked to enhanced cocaine reward, highlights the role of specific members of the gut microbiota in modulating addiction-related behaviors.

While both Kiraly et al. (2016) [9] and Cuesta et al. (2022) [10] found that antibiotics can enhance CPP or cocaine-induced behaviors, Kiraly et al. observed a dose-dependent effect, with a lower dose enhancing CPP. In contrast, the current study suggests that microbiome depletion, whether by antibiotics or GF conditions, attenuates CPP responses. In GF mice, microbiome restoration restored CPP to control levels, further emphasizing the importance of the gut microbiome in modulating cocaine reward behavior. Thus, the current study demonstrates the critical role of a functional microbiome for normal CPP responses, aligning with previous studies but highlighting differing interpretations regarding the effects of antibiotics on CPP.

### 4.5. Striatal Levels of DAT and TH

The role of the gut microbiota in influencing dopaminergic signaling and its implications for neurological and metabolic disorders has gained significant attention in recent years [34]. However, the results from this study suggest that alterations in the gut microbiome, whether by acquisition or depletion, do not appear to directly affect dopaminergic signaling, at least under the conditions studied. Our findings showed no significant changes in DAT or TH levels between GF and CONV-R mice (Figure 2A). Similarly, the antibiotic-treated mice did not exhibit significant differences in DAT or TH levels when compared to controls (Figure 2B). Additionally, our study revealed significant sex differences in DAT and TH levels, with females showing generally higher DAT and higher TH levels compared to males, regardless of microbiome status (Figure 2C,D). These findings suggest that sex hormones or other biological factors may modulate dopaminergic system activity independently of the microbiome and highlight the importance of considering sex as a biological variable in studies of dopaminergic signaling. Further investigation is needed to explore how sex-related factors interact with the microbiome and influence the dopaminergic system.

### 4.6. Striatal DA Levels and Turnover Rate

HPLC analysis of DA and its metabolite DOPAC indicated that microbial status significantly influenced both DA and DOPAC levels in the GF model. In particular, the acquisition of a gut microbiome (CONV) resulted in a significant increase in DA levels in GF mice, while DA levels in the CONV-R mice remained stable (Figure 3A). Initially, DOPAC levels in GF mice were higher than in CONV-R mice, but after conventionalization, DOPAC levels in GF mice decreased significantly, becoming comparable to those in CONV-R mice. Moreover, DOPAC/DA ratios showed a significant increase in GF mice at baseline, which was normalized after conventionalization (Figure 3C).

These results align with the findings of Heijtz et al. [35], who found elevated DA turnover in GF NMRI male mice compared to SPF mice, indicated by a higher DOPAC/DA ratio, suggesting increased DA metabolism. While Heijtz et al. did not investigate the effects of introducing a conventional microbiome, our study extends their work by showing that transitioning from a GF to a conventional microbiome normalizes DA and DOPAC levels, suggesting the gut microbiome modulates DA turnover.

Our study also shows that microbiome-induced changes in DA turnover can be reversed following conventionalization, supporting the gut–brain axis as a key mediator of neurotransmitter regulation. The normalization of DA and DOPAC levels after conventionalization parallels the elevated DA turnover observed in GF mice by Heijtz et al., further supporting the role of the gut microbiome in modulating DA metabolism across different conditions.

In contrast, the ABX model showed no significant differences in DA, DOPAC, or DOPAC/DA ratios between control and ABX mice, although significant sex differences were found, with females showing higher DOPAC levels and DOPAC/DA turnover compared to males (Figure 3B,D,G). These findings highlight the impact of gut microbiome status on dopaminergic function in the GF model, while sex differences in dopamine metabolism should be considered in both GF and ABX models. The lack of significant differences in the ABX model may be due to antibiotic-resistant microbes maintaining a stable microbiota composition, preventing alterations in dopaminergic activity.

In the current study, normalization of striatal DA activity and turnover following repletion of the gut microbiota in GF mice supports the hypothesis that the gut microbiome plays a role in modulating DA metabolism and addiction behaviors. Dysregulation of dopamine activity, particularly in the mesolimbic and mesocortical systems, is closely linked to anxiety [36]. Such altered dopaminergic signaling can contribute to heightened stress responses and a tendency to seek substances that modulate these pathways for relief [37]. Furthermore, the co-occurrence of anxiety disorders and substance use disorders (SUDs) has been well-documented [38,39]. Additionally, previous studies have shown that alterations in gut microbiota influence dopaminergic neurotransmission [40], illustrating the microbiome’s potential involvement in anxiety and SUDs. Together, these findings indicate that the gut microbiome not only plays a key role in modulating dopaminergic function but also contributes to the complex interplay between anxiety and substance use disorders, opening new avenues for potential therapeutic strategies.

### 4.7. Bacterial Load

In the GF model, the lack of significant differences in bacterial load suggests that the conventional microbiome successfully colonized the GF mice, restoring a typical microbial profile (Figure 4A). This indicates that conventionalization effectively reestablished a normal gut microbiota without substantial disruptions.

In the ABX model, the absence of significant differences in bacterial load after 18 days of treatment (Figure 4G) suggests that despite antibiotic exposure, the gut microbiome was recolonized with antibiotic-resistant bacteria. This recolonization may allow for the recovery of certain microbiota functions and could mask potential impacts that would have been observed with more complete eradication. The persistence of resistant bacteria may help maintain a stable microbial environment despite antibiotic treatment.

### 4.8. Microbial Alpha and Beta Diversity

In the GF model, significant differences in alpha and beta diversity were observed between CONV-R GF and CONV-R mice (Figure 4B–D). Differential abundance analysis showed these differences were mainly due to changes in Lachnospiraceae, *Akkermansia*, and Muribaculaceae ASVs, with *Akkermansia* and Muribaculaceae more abundant in GF mice at the expense of Lachnospiraceae (Figure 5A). Despite these taxonomic differences, the gut microbiota of conventionalized GF mice was more similar to that of CONV-R mice than antibiotic-treated mice. These findings suggest that conventionalization led to substantial changes in microbiome composition, though full restoration of diversity was not achieved within this study’s timeframe.

In the ABX model, significant reductions in alpha diversity were observed in ABX mice compared to controls (Figure 4H–J). Differential abundance analysis revealed substantial changes, including reductions in Lachnospiraceae ASVs and an increase in a single Enterobacteriaceae ASV (Figure 5B). BLAST analysis identified this increased Enterobacteriaceae ASV as being identical to *Enterobacter cloacae* and *Klebsiella* spp., both of which are resistant to many antibiotics [41,42]. Beta diversity analysis confirmed significant differences between ABX and control groups, indicating that antibiotic treatment induced shifts in microbial community structure, contributing to antibiotic resistance. These findings align with Cuesta et al., 2022 [10], who observed that antibiotics disrupt gut microbiota, reducing its diversity and enabling colonization by γ-Proteobacteria, which can lead to significant metabolic and behavioral changes in the host. These reductions in microbial diversity and shifts in community composition are consistent with observations in the GF model, where the absence of microbiota similarly impacts host physiology and behavior, emphasizing the gut microbiome’s role in modulating behavioral responses such as those to cocaine.

### 4.9. Inferred Microbial Metabolic Pathway Abundance

The inferred microbial metabolic pathway analysis highlighted contrasting functional impacts between the two models. In the GF model, conventionalization resulted in minor metabolic changes, such as reduced methanogenesis and L-lysine fermentation, suggesting limited functional disruption during microbiome establishment (Figure 6A,B). In the ABX model, however, substantial alterations were observed, including increased degradation of GABA and decreased production of SCFAs, indicating disruptions in neurotransmitter and energy-related pathways (Figure 6C,D). Specifically, we observed the downregulation of succinate fermentation to butanoate and acetyl-CoA fermentation to butanoate, both crucial for butyrate synthesis. Additionally, enhanced siderophore biosynthesis and shifts toward oxidative metabolism reflected microbial adaptation to antibiotic stress [43]. These results highlight the stark contrast between the profound functional disruptions induced by the antibiotic treatment and the modest changes observed with conventionalization in GF mice.

SCFAs produced by gut microbes during the fermentation of dietary fibers influence brain function and behavior through several mechanisms, including modulation of the blood-brain barrier, neuroinflammatory pathways, and neurotransmitter levels [44,45]. Although SCFA levels were not directly measured in this study, the microbiome-associated alterations in inferred fatty acid biosynthesis in ABX mice suggest that microbiome-driven changes in SCFA production may play a role in the observed effects on cocaine reward and locomotion. This hypothesis aligns with the finding of Meckel et al. [8], which indicated that repletion of SCFAs reversed the effects of antibiotics on cue-induced cocaine-seeking after abstinence, normalizing active lever pressing behavior to control levels. This finding potentially links altered microbial metabolism to the behavioral changes observed in antibiotic-treated mice in response to cocaine.

Meckel et al. (2024) investigated the effect of antibiotic-induced microbiome depletion on cocaine self-administration and drug-seeking behavior after abstinence in rats [8]. They found that while microbiome depletion did not alter the acquisition of cocaine self-administration on a fixed-ratio schedule, it significantly increased motivation for low-dose cocaine in a within-session threshold task. Additionally, microbiome-depleted rats displayed heightened drug-seeking behavior in response to a drug-paired cue after 3 weeks of forced abstinence. These effects were reversed by the repletion of SCFAs, suggesting a potential mechanism for the observed behavioral changes. Although the in-session observations in rats by Meckel et al. (2024) [8] differ from the current study’s findings in mice, both studies indicate the reduction in SCFAs as a mechanism mediating the microbiome’s actions on cocaine responses.

Similar to the research by Hofford et al. (2021) on morphine reward [46], our findings suggest that changes in SCFA production, driven by gut microbiome alterations, may play a significant role in influencing drug-related behaviors, including those associated with cocaine. In their study, antibiotic-induced microbiome depletion reduced microbial diversity and the relative abundance of Firmicutes, leading to a diminished response to morphine, evidenced by blunted locomotor sensitization and CPP. These behavioral changes were reversed upon SCFA supplementation, underscoring the critical role of SCFAs in modulating drug reward. Similarly, our study’s microbiome-driven alterations in inferred fatty acid biosynthesis suggest that changes in SCFA production may also contribute to the observed effects on cocaine reward and locomotion. This influence likely occurs through their impact on brain regions involved in the reward system, possibly via epigenetic or other molecular mechanisms, offering new insights into potential interventions for addiction.

Additionally, in the current study, the inferred increase in microbial degradation of GABA in ABX mice suggests potential effects on circulating neurotransmitter levels, influencing behavior. GABA is a key inhibitory neurotransmitter, and its altered metabolism may contribute to the enhanced behavioral responses to cocaine observed in our study. Previous research has shown that GABAergic modulation, particularly through GABAB agonists, can reduce cocaine self-administration (Roberts & Brebner, 2006) [47]. Furthermore, the inferred functional analysis revealed significant changes in numerous bacterial metabolic pathways due to antibiotic treatment, highlighting the potential role of the microbiome and its metabolites, including GABA, in modulating cocaine-seeking behaviors. These findings lay the foundation for future translational work in this area.

Supporting this, Tran et al. (2023) [32] reported strain-specific significant differences in orthologous genes related to GABA/glutamine/glutamate and SCFA metabolisms in cocaine-sensitized male mice. Orthologous genes such as K11102 (other_glutamate) and K13923 (propionate synthesis) showed the largest fold changes. Their analysis also revealed significant differential expression in pathways like glutamate degradation, purine nucleotide degradation, and DA degradation post-cocaine sensitization. These findings underscore the complex interplay between the host genome, gut microbiome composition, metabolic pathways, and behavior, highlighting the potential for microbiome-targeted interventions in modulating neurobehavioral outcomes.

The results of this study have important implications for addiction research and treatment. They highlight the gut microbiome as a key modulator of drug reward and addiction, suggesting that interventions aimed at restoring or manipulating the microbiome may have therapeutic potential. For example, dietary interventions, probiotics, or microbiome-targeted therapies could be explored as adjuncts to traditional addiction treatments. Furthermore, the findings underscore the need for further research to elucidate the specific microbial taxa, metabolic pathways, and host factors that contribute to these effects and to determine how these factors interact to modulate drug-induced behaviors.

## 5. Conclusions

In conclusion, our study investigates the relationship between the gut microbiome and cocaine-seeking behavior using GF and ABX animal models, along with 16S rRNA gene analysis. Our findings highlight the complex interplay between the gut microbiota, DA neurochemistry, and metabolic shifts. This research enhances our understanding of how microbiome depletion impacts brain function and behavior in the context of addiction. Further investigation into the mechanisms by which the microbiome influences drug-seeking behavior is warranted and could reveal valuable insights for identifying therapeutic targets for addiction-related disorders.

## 6. Limitations

While our study provides valuable insights into the role of the gut microbiome in cocaine-seeking behavior, several limitations should be noted. First, the analysis was conducted at specific time points, which, while allowing for a focused examination of microbiome and behavioral changes at critical stages, may not capture the full range of changes over longer durations. Future studies could benefit from incorporating extended time points to explore potential long-term effects. Additionally, our use of germ-free and antibiotic-treated animal models, while providing a controlled environment to study the direct effects of microbiome depletion, may not fully represent the complexity of human microbiota, limiting the generalizability of our findings. Furthermore, our functional pathway analysis relied on inferred PICRUSt2 predictions. While this method is useful for predicting microbial functions based on 16S rRNA data, it depends on available reference databases for gene classification and may not capture the full spectrum of pathways or strain-specific functions. Lastly, this study focused on a limited range of behavioral outcomes, which allowed for a more in-depth analysis of these specific behaviors. However, expanding the range of behavioral assessments could provide a more comprehensive understanding of the microbiome’s role in addiction-related behaviors.

## Figures and Tables

**Figure 1 microorganisms-13-00077-f001:**
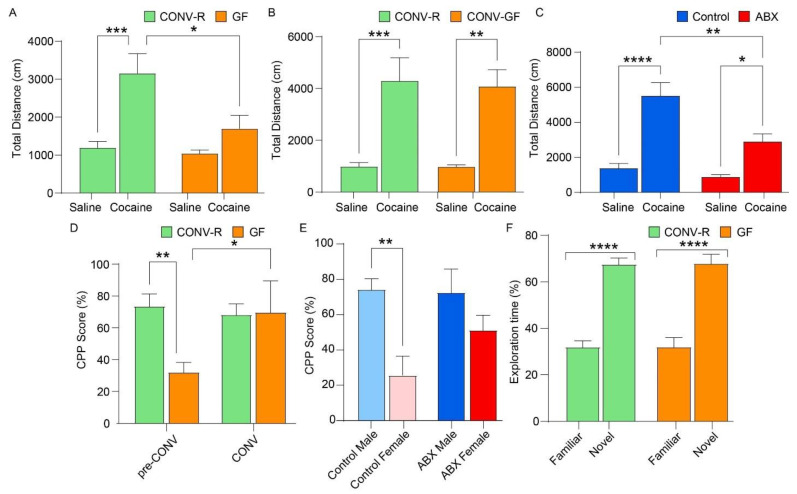
Psychomotor activation, cocaine-conditioned place preference (CPP), and exploratory behavior in germ-free (GF) and antibiotic (ABX) models: (**A**) bar graph showing locomotor activity in GF and CONV-R mice following cocaine treatment prior to conventionalization; (**B**) bar graph showing locomotor activity comparing GF and CONV-R mice after cocaine treatment after conventionalization; (**C**) bar graph showing locomotor responses in ABX-treated mice following cocaine treatment; (**D**) bar graph showing the acquisition of CPP in GF and CONV-R mice; (**E**) bar graph showing CPP scores in ABX mice stratified by sex; and (**F**) bar graph showing performance in the Novel Object Recognition (NOR) test in GF and CONV-R mice. Significance is indicated as follows: * *p* < 0.05, ** *p* < 0.01, *** *p* < 0.001, and **** *p* < 0.0001.

**Figure 2 microorganisms-13-00077-f002:**
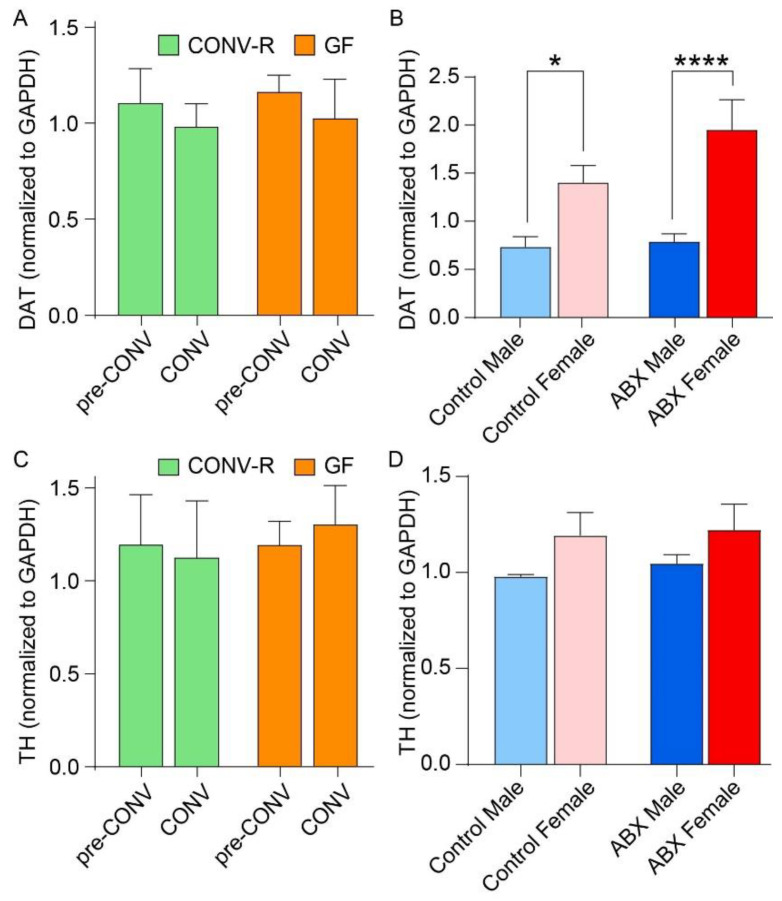
Striatal levels of tyrosine hydroxylase (TH) and dopamine transporter (DAT) in germ-free (GF) and antibiotic (ABX) models: (**A**) bar graph showing striatal DAT levels in GF mice; (**B**) bar graph showing striatal DAT levels in the ABX model, comparing control and ABX-treated mice, stratified by sex; (**C**) bar graph showing striatal TH levels in GF mice; and (**D**) bar graph showing striatal TH levels in the ABX model. Significance is indicated as follows: * *p* < 0.05 and **** *p* < 0.0001.

**Figure 3 microorganisms-13-00077-f003:**
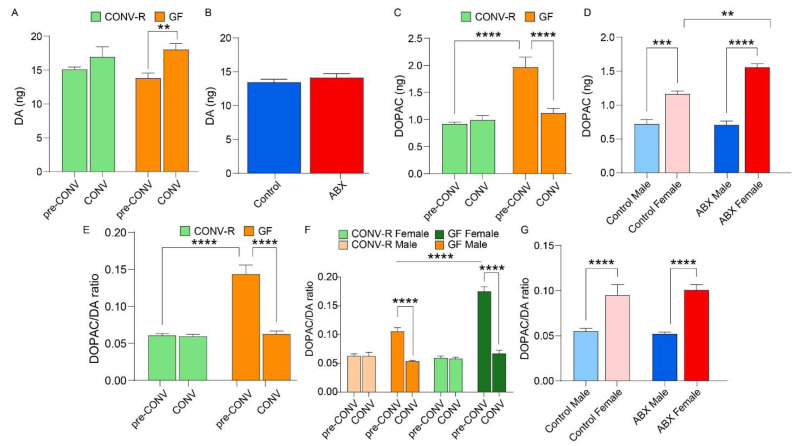
Striatal dopamine (DA) levels and turnover rate in germ-free (GF) and antibiotic (ABX) models: (**A**) bar graph showing DA levels in the GF model; (**B**) bar graph showing DA levels in the ABX model; (**C**) bar graph showing DOPAC levels in the GF model; (**D**) bar graph showing DOPAC levels in the ABX model; (**E**) bar graph showing DOPAC/DA ratio in the GF model; (**F**) bar graph showing DOPAC/DA ratio in the GF model, stratified by sex; and (**G**) bar graph showing DOPAC/DA ratio in the ABX model, stratified by sex. Significance is indicated as follows: ** *p* < 0.01, *** *p* < 0.001, and **** *p* < 0.0001.

**Figure 4 microorganisms-13-00077-f004:**
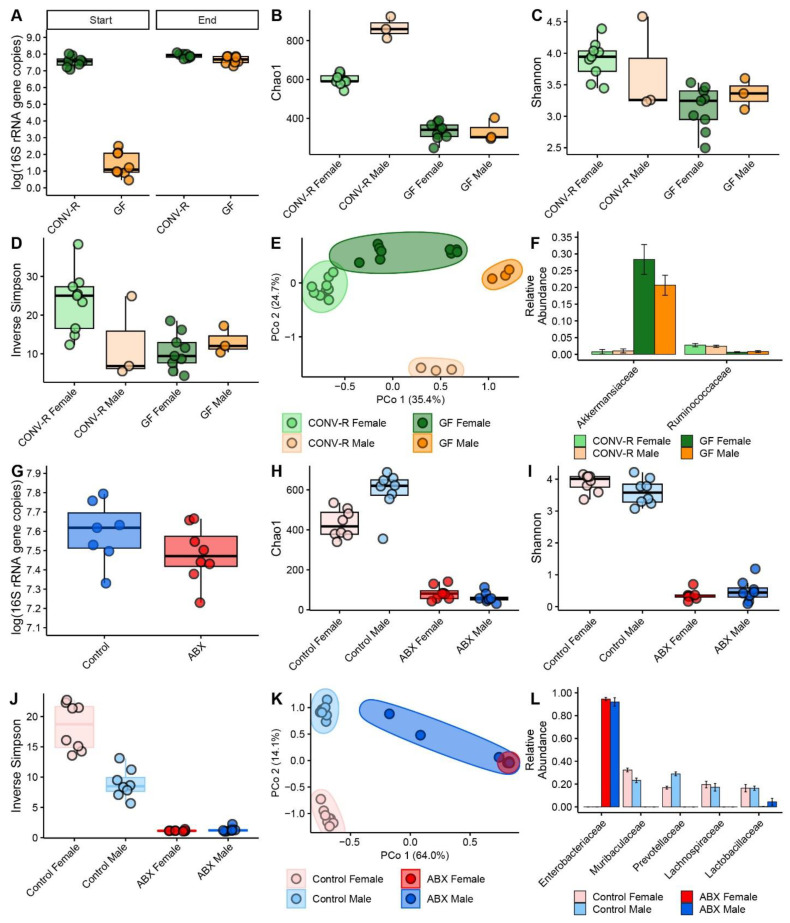
Alpha diversity metrics in GF and ABX models: (**A**) bar graph showing bacterial load measured by qPCR in the GF model; (**B**) bar graph showing Chao1 index values in the GF model; (**C**) bar graph showing Shannon index values in the GF model; (**D**) bar graph showing inverse Simpson index values in the GF model; (**E**) principal coordinates analysis (PCoA) depicting the variation in structure (Bray–Curtis Index) of fecal 16S rRNA gene profiles in the GF model; (**F**) bar graph showing differentially abundant bacterial taxa at the Family-level (Rank 6) in the GF model; (**G**) bar graph showing bacterial load measured by qPCR in the ABX model; (**H**) bar graph showing Chao1 index values in the ABX model; (**I**) bar graph showing Shannon index values in the ABX model; (**J**) bar graph showing inverse Simpson index values in the ABX model; (**K**) PCoA depicting the variation in structure (Bray–Curtis Index) of fecal 16S rRNA gene profiles in the ABX model; and (**L**) bar graph showing differentially abundant bacterial taxa at the Family-level (Rank 6) in the ABX model.

**Figure 5 microorganisms-13-00077-f005:**
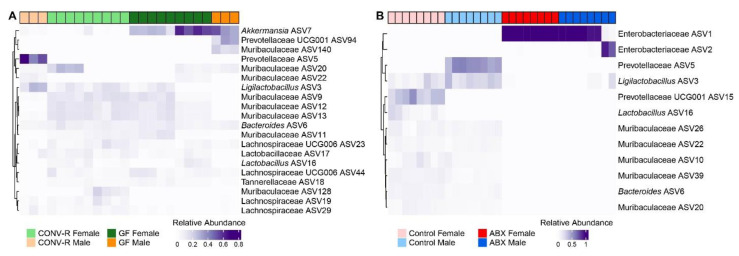
Heatmap illustrating the relative abundance of bacterial ASVs (amplicon sequence variants) in the fecal 16S rRNA gene profiles of female and male mice in the (**A**) GF (germ-free) and (**B**) ABX (antibiotic-treated) models. The heatmap displays differences in microbial composition across treatment groups, with color intensity reflecting the relative abundance of ASVs. Clustering of rows and columns was performed using Bray–Curtis distance to assess similarities in microbial communities across all ASVs and within each treatment group. Data are stratified by sex to explore potential sex-based differences in microbial composition. Statistical comparisons were made to identify significant variations in ASV abundance between groups, emphasizing the role of the gut microbiome in shaping the experimental conditions.

**Figure 6 microorganisms-13-00077-f006:**
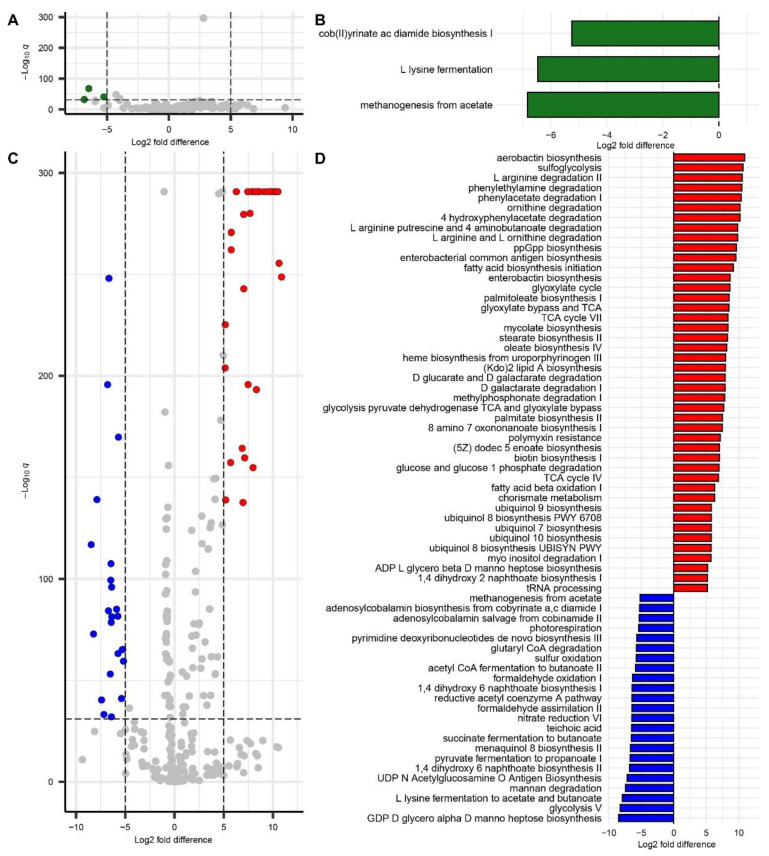
Differential abundance of inferred microbial pathways in GF and ABX models: (**A**) volcano plot showing the differential abundance of microbial pathways between germ-free (GF) and conventionally housed (CONV) mice; (**B**) differential abundance plot highlighting pathways significantly altered in the GF model; (**C**) volcano plot displaying the differential abundance of microbial pathways between antibiotic-treated (ABX) and control mice; and (**D**) Differential abundance plot illustrating pathways significantly altered in the ABX model. In all plots, microbial pathways were inferred using PICRUSt2, and differential abundance was assessed using the Maaslin2 analysis with a *q*-value cutoff of 10^−32^ and a fold change threshold of 5.0. Pathways with higher abundance in CONV mice compared to GF mice are shown in green, those more abundant in ABX mice are in red, and those more abundant in control mice are in blue. Pathways shown in gray represent those that did not meet both the *q*-value and fold change thresholds.

## Data Availability

The 16S rRNA gene sequencing files for fecal samples have been deposited in the National Center for Biotechnology Information Sequence Read Archive (SRA) as BioProject PRJNA1205714.

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
