# Peer review of "The Gut Microbiome Regulates the Psychomotor Effects and Context-Dependent Rewarding Responses to Cocaine in Germ-Free and Antibiotic-Treated Animal Models"

_microorganisms, 2025, doi:10.3390/microorganisms13010077_

Round 1
Reviewer 1 Report
Comments and Suggestions for Authors
The article discusses the role of the gut microbiome in regulating the psychomotor effects of cocaine in the mouse model. The rationale is adequate, and the technical methods are satisfactory. Perhaps the only detail would have been to pursue a more extended analysis time point. The scheme presented in Figure 1 is perfect and clear. Figures 2, 3, and 4 give insight into the differences in treatment and gender. The legends of Figures 6 and 7 should be more informative. The discussion is well-defined and easy to follow but a little bit long. A similar point can be observed in the conclusions. At the end the limitations of the study should be describe.
Author Response
Comment 1: Perhaps the only detail would have been to pursue a more extended analysis time point.
Response 1: We appreciate this suggestion. The current time points were selected based on the study's focus and existing literature, which indicate that these time points are most informative for the measured outcomes. We acknowledge that extending the analysis could provide additional insights and will consider this in future work. This has also been noted in the limitations section.
Comment 2: The legends of Figures 6 and 7 should be more informative.
Response 2: We have revised the legends for Figures 6 and 7 (now Figures 5 and 6) to provide additional context and clarity, ensuring that the information is more comprehensive and accessible for readers.
Comment 3: The discussion is well-defined and easy to follow but a little bit long.
Response 3: We have revised the discussion to streamline the content, removing redundancy while maintaining the essential points. This adjustment helps make the discussion more concise without compromising the depth of the analysis.
Comment 4: A similar point can be observed in the conclusions.
Response 4: The conclusion has been revised to make it more concise while still effectively summarizing the key findings and implications of the study.
Comment 5: At the end the limitations of the study should be described.
Response 5: We have added a limitations section at the end of the manuscript, addressing the key limitations of the study and suggestions for future research.
Reviewer 2 Report
Comments and Suggestions for Authors
1. Add study design to the title
2. Abstract must be shortened and more precise, add numerical results
3. Line 37 - explain in depth peripheral effects
4. Introduction could have 2 paragraphs, one on cocaine and the other on microbiome, more information could be added to both. Also please add aim at the end
5. Delete Figure 1 or add as supplementary material
6. Nicely written methods section
7. Other figures are appropriate
8. References are up to date
9. Please add limitation section
Interesting study which adds to the body of literature in the addiction and microbiome connection. Few suggestions above would improve the manuscript
Author Response
Comment 1: Add study design to the title.
Response 1: We appreciate this suggestion. We have updated the title to include the study design, as suggested, to better reflect the approach used in the study.
Comment 2: Abstract must be shortened and more precise, add numerical results.
Response 2: We have shortened the abstract and made it more precise, incorporating relevant numerical results to provide a clearer overview of the study’s findings.
Comment 3: Line 37 - explain in depth peripheral effects.
Response 3: We have expanded the explanation of peripheral effects in the manuscript to provide a more comprehensive understanding of their role in the study.
Comment 4: Introduction could have 2 paragraphs, one on cocaine and the other on microbiome, more information could be added to both. Also, please add aim at the end.
Response 4: We have revised the introduction to include three distinct paragraphs: the first focusing on cocaine and its addictive properties, the second on the microbiome and its role in drug-related behaviors, and the third providing more detailed background on the use of germ-free and antibiotic-treated models in the study of addiction. Additionally, we have added a clear aim at the end of the introduction, outlining the purpose of the study to explore psychomotor and context-dependent rewarding responses to cocaine using both germ-free (GF) and antibiotic-treated (AF) models.
Comment 5: Delete Figure 1 or add as supplementary material.
Response 5: We have removed Figure 1 from the main manuscript and included it as supplementary material, as suggested, to streamline the presentation of the results.
Comment 6: Please add limitation section.
Response 6: A limitations section has been added to the manuscript, addressing the key limitations of the study and outlining areas for future research.